# Emerging Roles of RAD52 in Genome Maintenance

**DOI:** 10.3390/cancers11071038

**Published:** 2019-07-23

**Authors:** Manisha Jalan, Kyrie S. Olsen, Simon N. Powell

**Affiliations:** Department of Radiation Oncology, Memorial Sloan Kettering Cancer Center, New York, NY 10065, USA

**Keywords:** RAD52, RNA:DNA hybrids, R-loops, DNA repair, double-strand break repair, BRCA1, BRCA2, RAD51, synthetic lethality, genome instability

## Abstract

The maintenance of genome integrity is critical for cell survival. Homologous recombination (HR) is considered the major error-free repair pathway in combatting endogenously generated double-stranded lesions in DNA. Nevertheless, a number of alternative repair pathways have been described as protectors of genome stability, especially in HR-deficient cells. One of the factors that appears to have a role in many of these pathways is human RAD52, a DNA repair protein that was previously considered to be dispensable due to a lack of an observable phenotype in knock-out mice. In later studies, RAD52 deficiency has been shown to be synthetically lethal with defects in BRCA genes, making RAD52 an attractive therapeutic target, particularly in the context of BRCA-deficient tumors.

## 1. Double-Strand Breaks and RAD52

Double-strand breaks (DSBs) are the greatest threat to genomic integrity and can result from endogenous sources such as reactive oxygen species (ROS) and replication stress or exogenous insults such as UV, ionizing radiation (IR), and chemotherapeutic compounds. If left unrepaired, DSBs can cause chromosomal aberrations, mutagenesis, and cell death, leading to the development of severe diseases including cancer [1,2]. Given the cytotoxic nature of these lesions, the cell has developed multiple pathways of DSB repair, some which result in mutagenic repair, and others that use a template to ensure error-free repair.

The majority of DSBs in a cell are repaired by non-homologous end-joining (NHEJ), which ligates blunt ends in a sequence-independent manner and can result in small insertions or deletions (indels) [1]. Homologous recombination (HR) is generally considered the most faithful DSB repair pathway, as it uses the intact sister chromatid as a donor, and it is highly dependent on breast cancer associated genes BRCA1 and BRCA2. The usage of a sister chromatid means that HR is predominately active in S or G2 when the cell has replicated its DNA [3,4]. In order to initiate HR, the DSB is resected, leaving a segment of single-stranded DNA (ssDNA) that is coated first by RPA, then by RAD51 in order to facilitate homology search and invasion of the sister chromatid—a process known as strand exchange. BRCA1 is predominately involved in the resection step of HR, while BRCA2 mediates removal of RPA and loading of RAD51; however, recent work has shown that BRCA1 may also have a function in the later steps of HR [3,5,6].

While NHEJ is the predominant pathway for DSB repair, its requirement for blunt ends renders it incapable of dealing with resected ends or one-ended DSBs resulting from replication fork collapse. Replication stress is the most common source of endogenous damage and is usually repaired by HR proteins [7,8]. In an HR-deficient context, DSBs that cannot be repaired by NHEJ must be channeled through error-prone homology-mediated pathways such as single-strand annealing (SSA) and alternative end-joining (Alt-EJ). In SSA, long-range resection proceeds bi-directionally from the DSB until homologous sequences are revealed. The break is then repaired by the annealing of these complementary sequences, resulting in deletion of the intervening sequence [3,9]. Alt-EJ operates by an analogous mechanism, only it requires a shorter length of resection and anneals regions of microhomology, resulting in smaller deletions. While the pattern of resection and annealing of complementary sequences are similar, Alt-EJ and SSA have been shown to require different protein factors [3,9,10]. SSA has canonically been considered the only RAD52-dependent pathway, as it is responsible for annealing the resected ends [9,11,12,13,14]; however, recent work by the Hendrickson lab using previously established plasmid reporter assays has shown that HR, Alt-EJ, and SSA all decrease in a RAD52-null cell [15].

Rad52 (Radiation sensitive 52) was first identified in budding yeast in a screen for mutations that would inhibit repair of IR-induced DSBs, resulting in radio-sensitization [16,17]. Several other DNA repair proteins were also identified in this screen and are referred to as the Rad52 epistasis group. Of this group, most proteins were found to play similar roles in yeast and human cells, while Rad52 had a marked difference between its functionality in yeast and in higher eukaryotes [14].

While mammalian cells rely on BRCA2 to mediate loading of RAD51 onto RPA-coated ssDNA and on RAD52 for annealing ssDNA ends, in yeast, both of these functions are performed by Rad52, as they lack a functional homolog of BRCA2 [18]. Consequently, Rad52-mutant yeast show defects in meiotic recombination, spore viability, HR, and mating-type switching [16], while RAD52-null mice display no observable phenotype in viability or fertility and only a mild effect on HR [19]. Despite the apparent dispensability of RAD52 in mammals, one of the initial hypotheses for a function of RAD52 in human cells came from our lab’s work showing that BRCA2-deficient cells required RAD52 for survival, indicating a synthetically lethal relationship [20]. In a BRCA-deficient context, RAD51 loading can be carried out by RAD52, albeit at a lower efficiency [20,21]. A recent study in vitro has supported this function of RAD52 by using DNA curtain analysis to show the dynamics of interactions between RPA, RAD52, RAD51, and ssDNA [22]. RAD52′s potential involvement in HR was also supported by the finding that GFP-tagged RAD52 formed foci in response to DNA damage throughout S and G2, coinciding with HR activity during these stages of the cell cycle [23]. Since then, RAD52 has been implicated in many other pathways of DNA repair and genome maintenance, most of which are also dominant in S and G2 (Figure 1) [24,25,26,27,28,29,30].

## 2. Biochemistry and Activity of RAD52

Human RAD52 consists of a C-terminal domain (CTD) which contains its nuclear localization signal (NLS) as well as its RAD51- and RPA-binding regions, and an N-terminal domain (NTD) which has been shown to be responsible for its multimerization and DNA-binding activity [16,31,32,33,34,35]. RAD52 can also undergo context-dependent post-translational modifications such as SUMOylation, phosphorylation, acetylation, and methylation, the functions of which are under active investigation [16,36,37,38,39,40].

Full-length RAD52 exists as a heptameric ring, while purified NTDs have been shown to form an undecameric ring structure [41,42]. While a ring structure would usually be suggestive of a helicase, the DNA-binding domain of RAD52 is a groove of positively-charged residues which wraps around the outside of the ring, unlike helicases, which typically thread DNA through the center of the ring [32,41,43]. This domain has been shown to bind DNA with a footprint of 4 nucleotides per monomer, meaning that each heptameric ring could bind around 28 bp of DNA [43,44,45]. RAD52 has also been shown to contain a secondary DNA-binding site that can interact with ssDNA, which would facilitate annealing of resected ends, as in SSA. This site can also bind double-stranded DNA (dsDNA), which enables strand exchange in an ATP-independent manner [31,46]. Recent studies using atomic force spectroscopy have provided insights into the mechanics of RAD52/DNA interactions and illuminated possible molecular mechanisms for RAD52-facilitated annealing and strand exchange [47]. In addition to canonical or “forward” strand exchange, where RAD52 binds ssDNA and facilitates its invasion of dsDNA, RAD52 can also perform inverse strand exchange by binding the duplex DNA, which is then invaded by ssDNA or RNA [48,49]. 

## 3. Replication-Associated Functions of RAD52

RAD52 plays an important role in repair processes associated with replication stress, and, unlike HR, its involvement in these pathways is largely RAD51- and BRCA2-independent [24,25,26,27,28,29]. At stalled forks, RAD52 helps maintain equilibrium by preventing excessive fork reversal and stabilizing the fork. It converts the stalled fork into a compact conformation, making it less accessible to fork reversal enzymes [27]. In a CHK1-deficient background, RAD52 can convert reversed forks into a structure which is then cleaved by MUS81 and repaired by break-induced replication (BIR) [24]. BIR occurs when a broken chromosome invades its homologous duplex and replication proceeds along the length of the chromosome, resulting in conservative replication, and it was shown in budding yeast to be dependent on Rad52 [14]. Since then, human RAD52 has also been shown to be a requirement for BIR at DSBs resulting from fork collapse [26], as well as a subset of microhomology-mediated BIR known as MiDAS (mitotic DNA synthesis) [25]. MiDAS ensures replication of common fragile sites and other difficult to replicate regions in mitosis to ensure complete replication before cell division [25,29,50,51]. DNA that evades replication by MiDAS is known as under-replicated DNA (UR-DNA) and can form ultra-fine anaphase bridges that result in 53BP1 nuclear bodies in the daughter cells. Within these nuclear bodies, there is a “second chance” mechanism, also dependent on RAD52, for this UR-DNA which allows the cell to complete replication faithfully [28]. RAD52 also helps ensure telomere stability through multiple pathways for alternative lengthening of telomeres (ALT), as demonstrated by the shortening of telomeres over successive cell cycles in RAD52-deficient cells [29,30,50,51].

## 4. A New Interactome of RAD52: RNA:DNA Hybrids

### 4.1. RAD52′s Interactions with DNA and RNA

RAD52′s role in DNA repair has been attributed to its affinity for both single-stranded and double-stranded DNA. Nevertheless, recent studies show that RAD52 may have the same, if not even greater, affinity for RNA substrates (Figure 2A) [52]. This affinity allows RAD52 to perform many of its canonical functions using RNA in place of ssDNA. For instance, human RAD52 can facilitate inverse strand exchange between ssRNA and dsDNA, promoted in the presence of RPA [48,53]. Surprisingly, in humans, this process is not dependent on RAD51, while in both bacteria and yeast, inverse strand exchange with RNA requires their respective RAD51 homologs [48,49,54]. Further supporting its association with RNA, RAD52 has been found to interact with RNA polymerase II (RNAPII), indicating an association with nascent RNA transcripts [55].

Consistent with its role in HR, RAD52 has also been implicated in a newly described form of homology-directed repair where transcript RNA is used to repair double-strand breaks in the absence of a DNA donor, contradicting the central dogma of biology [53,56,57].

### 4.2. RAD52 and RNA-Templated Repair 

The first evidence that a double-strand break could be repaired using RNA as a template came from experiments done in budding yeast [56]. Because RNA is so abundant in cells, it was proposed that, in the absence of a sister chromatid, a homologous RNA sequence could function as a template for error-free repair of a double-strand break. This RNA could be derived from a nascent transcript on the same copy of the gene as the DSB (*cis*) or from another locus (*trans*) [53,58]. The Storici lab developed a splicing-based system to ensure that they could distinguish DSB repair that had used an RNA template from repair that had undergone conventional HR. They also looked at yeast retrotransposons to determine whether RNA-templated repair occurred directly through an RNA donor or whether the RNA was reverse transcribed into a cDNA intermediate. They found that in an RNaseH-deficient background, RNA could be directly used as a template even in the absence of retrotransposons, indicating that in this context, not all repair events required a cDNA intermediate. Additionally, knocking down Rad52 reduced the frequency of RNA-templated repair in *cis* [48,53]. Although its frequency is low and its relevance is context-dependent, this novel pathway of DNA repair was shown to be conserved across species from bacteria to humans [59,60].

While RNA-templated repair was initially described using yeast Rad52, later work by the Pomerantz lab has implicated human RAD52 in RNA-templated repair in vitro. They showed that RNA may participate in DSB repair either as a scaffold to assist in ligation of the broken ends (bridging) or as a template providing the necessary sequence information for homologous repair (RNA-templated DSB repair). Both mechanisms are facilitated by human RPA and RAD52, and form a hybrid structure known as an R-loop [57].

### 4.3. R-Loops: Physiological RNA:DNA Hybrids

An R-loop is a three-stranded structure formed when RNA invades duplex DNA, displacing one of the DNA strands. Physiologically, this structure is most commonly formed during transcription when the RNAPII complex unwinds the dsDNA, displacing the non-template strand and enabling the nascent RNA to hybridize with the homologous template strand [61,62]. This RNA:DNA hybrid structure is also relevant in initiation of replication, termination of transcription, class-switch recombination, and protection of telomeres [61,63], and is a necessary intermediate in the process of RNA-templated repair described above. R-loops are commonly formed at sequences with repeat motifs and/or high GC content, because the displaced strand of DNA in these sequences is able to form secondary structures such as G-quadraplexes, thus stabilizing the R-loop [64,65].

It has been shown biochemically that the RNA:DNA hybrid structure is even more stable than the double-stranded DNA, warranting mechanisms for efficient resolution of these hybrids so that the two strands of DNA can re-anneal [66]. Across species, this is often accomplished by the enzymes RNaseH1 and RNaseH2, which selectively degrade the RNA component of RNA:DNA hybrids [67,68,69].

#### 4.3.1. R-Loops and RAD52: A Scaffold for DNA Repair Proteins

In *S. pombe*, the formation and resolution of R-loops at DSBs has been shown to be required for efficient repair by HR [70]. Transcription has also been shown to influence pathway choice in DSB repair in humans, as shown by CHIP-seq data indicating that transcriptionally active genes preferentially recruit HR factors compared to untranscribed genes [71]. These studies indicate that R-loops can function as signals for downstream recruitment of repair proteins.

This scaffold role of R-loops in protein recruitment has been extensively investigated by the Lan lab, who have characterized a repair pathway they refer to as transcription-coupled homologous recombination (TC-HR) [52,72,73]. In a system that allowed for modulation of both transcription and site-specific damage induction, they were able to show that R-loops formed upon active transcription initiated a novel signaling cascade. The R-loops were first bound by Cockayne Syndrome B (CSB) protein, which then recruited RAD52, which loaded RAD51, defining TC-HR as a RAD52-dependent, BRCA1/BRCA2-independent pathway [73]. Since their research was done in post-mitotic neurons or non-proliferating U20S cells, where a sister chromatid is unavailable, they proposed that the proteins recruited were conducting HR-like repair using RNA as a template, and that TC-HR might be most relevant in G0/G1 (Figure 1) [52,72,73]. However, they do not determine whether RNA is directly being used as a template or merely as a scaffold for protein recruitment.

A similar pathway, termed transcription-associated homologous recombination repair (TA-HRR), has also been shown to require R-loops and RAD52 for repair of DSBs in actively transcribed genes. In this system, RAD52 is required for the recruitment of XPG, which cleaves R-loops into a substrate with a ssDNA overhang, similar to a resected end, that can then be processed by HR [74]. They indicate that RAD52 is recruiting the downstream HR factors, but it is also possible that it is the structure left behind by XPG cleavage that is being recognized by the HR machinery. Interestingly, unlike TC-HR, which is active in G0/G1 and BRCA-independent, TA-HRR was shown to involve BRCA1 and occur in S/G2 (Figure 1). 

While both TC-HR and TA-HRR demonstrate the presence of RAD52 at R-loops, the existing evidence that RAD52 directly binds R-loops comes from in vitro studies [52,72,73,74]. Given that RAD52 has affinity for a variety of nucleic acid substrates (Figure 2A), it could interact with the R-loop in any of three ways: at the displaced ssDNA strand, the nascent RNA, or the RNA:DNA hybrid (Figure 2B). Furthermore, because of its strand exchange activity, it is possible that RAD52 is promoting the formation of R-loops at a double-strand break by either forward or inverse strand exchange (Figure 2C).

#### 4.3.2. R-Loops as Agents of DNA Damage

Although R-loops have been shown to aid in various types of DNA repair, in some contexts R-loops may instead be the source of DNA damage. Since R-loop-induced defects are most prevalent in cycling cells, R-loops are thought to cause damage predominately by acting as a physical block to the progression of the replication fork, as the transcriptome and replisome must traverse along the same template. These transcription-replication collisions (TRCs) can lead to fork stalling and potential fork collapse [62,75]. R-loops also leave the displaced strand of DNA vulnerable to damage, such as spontaneous deamination [76,77]. Additionally, if the sequence of the displaced ssDNA is conducive to secondary structure formation, it is more likely to trigger other pathways tasked with resolving non-B forms of DNA. These forms of damage can result in mutations to the underlying DNA template, which has often been referred to as transcription-associated mutagenesis (TAM) [61,62,78].

Collisions between transcription and replication machinery can occur either in a head-on manner, where replication and transcription machinery are converging, or co-directionally, where both are proceeding in the same direction, albeit likely at different speeds. In species from bacteria to humans, highly transcribed genes have been found to preferentially transcribe in the same direction as replication so as to minimize head-on collisions. This led many to believe that head-on collisions were more deleterious to the cell than co-directional collisions. While recent experimental evidence has shown a greater accumulation of persistent R-loops in head-on TRCs, it remains to be seen if this leads to a greater degree of damage or mutagenesis [75]. 

Accumulation of R-loops has also been observed in the absence of crucial DNA repair proteins such as FANCM, FANCD2, BRCA1, and BRCA2 [79,80,81,82]. Mutation or loss of these proteins has been linked to diseases such as Fanconi Anemia and cancer, but it has yet to be determined whether the onset or progression of these diseases is linked to the accumulation of R-loops [62,83].

While these repair proteins have been shown to help prevent R-loop-induced damage, RAD52, which operates in many of the same pathways, has not appeared in any studies examining damage-causing R-loops. Nevertheless, the fact that it can bind R-loops in vitro and localize to R-loops present at break sites may indicate that a role for RAD52 at R-loop-induced damage is plausible, though this has yet to be investigated.

#### 4.3.3. The R-Loop Interactome

Not every R-loop results in damage, so it remains unclear in which contexts they would be pathological, neutral, or physiological. Nevertheless, even physiological R-loops must be processed efficiently, and the cell has developed many ways of regulating and resolving R-loops. Determining which proteins interact directly with R-loops has been a topic of great interest, and recent research has identified an R-loop interactome that is distinct from both the mRNA and DNA interactomes [84,85]. As expected, this interactome includes RNA-DNA helicases such as Senetaxin (STX), Aquarius (AQR), and DHX9, as well as RNaseH1 and RNaseH2, all factors known to resolve R-loops [63,84,85,86]. While the DNA repair factors previously mentioned were not identified in these interactomes, it is worth noting that these screens were not conducted in the presence of damage [84,85]. This may indicate that proteins such as RAD52 and CSB only localize to R-loops in the context of damage. Additionally, some proteins found to be recruited to R-loops, such as BRCA2, have not been shown to bind the nucleic acid structure directly, but instead are recruited because of their interaction with RNAPII [80].

## 5. RAD52 as a Target for Cancer Therapy

Despite its importance in a variety of repair pathways, RAD52 has long appeared dispensable for cell viability except in the context of BRCA-deficiency. This makes it an attractive therapeutic target in cancer, as tumors with BRCA mutations would be susceptible to loss of RAD52, while normal tissues with an intact HR pathway should survive without it. This genetic relationship is known as synthetic lethality, where loss of function of each gene on its own still results in viable cells, while loss of both results in cell killing. Synthetic lethality is an attractive strategy for targeted therapy, as it capitalizes on natural genetic relationships for tumor-specific kill instead of damaging agents, which can cause toxicity to both healthy cells and tumor cells [21,87].

In the context of DNA repair, synthetic lethality can especially be exploited when tumor cells deficient in one repair pathway are “addicted” to a backup pathway. Loss of the backup pathway then leads to toxic intermediate structures and cell death. The most common example of pharmacological exploitation of synthetic lethality is poly-ADP-ribose polymerase (PARP) inhibition in HR-deficient cells. Since PARP can recognize single-stranded breaks (SSBs), inhibiting its function would leave unrepaired SSBs which could be converted to DSBs during replication. In HR-defective cells, these breaks would either be left unrepaired or would be channeled to error-prone repair processes. The first FDA-approved PARP inhibitor, olaparib, was authorized for treatment in a subset of BRCA-mutated cancers in 2014, followed by the development and approval of a new generation of inhibitors [87]. While these inhibitors have been effective in some contexts, they have also been shown to cause global toxicity, likely because of the role of PARP proteins in cellular processes other than DNA repair [88]. The effectiveness of PARP inhibitors supported the use of synthetic lethality as an indicator for therapeutic success, but their adverse effects have inspired some to look to other synthetically lethal interactions for the treatment of HR-deficient tumors.

As homozygous loss of RAD52 caused no effect in mice, it is unlikely to cause toxicity to healthy tissues in humans. Its effect in HR-deficient tumors, on the other hand, is significant and well-characterized. While initially described as being synthetically lethal with BRCA2, it has since been found to also lead to cell death in the absence of BRCA1, PALB2, and RAD51 paralogs [20,21,89]. Additionally, recent super resolution microscopy data that mapped the spatiotemporal kinetics of HR repair showed that RAD52 and BRCA2 have distinct functions in the RAD51 nucleofilament formation and subsequent homology search. While in the absence of RAD52, BRCA2 was able to compensate for RAD52, the converse was not true [90]. This may explain why RAD52 loss in a BRCA2-proficient context has no observable effect. In the context of BRCA1 loss, one proposed mechanism for synthetic lethality with RAD52 was reliant on the endonuclease EEPD1, which recognizes stalled forks and cleaves them into a DSB. In the absence of BRCA1 and RAD52, accumulation of unrepaired breaks at collapsed forks, which are normally repaired by HR, results in cell death. The synthetic lethality is rescued by EEPD1 co-depletion, as the forks are restarted or repaired by other mechanisms which, while error-prone, do not involve these toxic intermediates [91]. 

Because RAD52 loss is so detrimental to BRCA-deficient cells, many groups have conducted screens to identify compounds that could serve as inhibitors of RAD52 function. The inhibitors target RAD52 in one of two ways: either by inhibiting its annealing activity by competitively binding to the ssDNA-binding domain, or by interfering with its ability to multimerize and form the ring structure required for its activity [87,92,93,94,95,96]. Some of these compounds may be useful in HR-deficient cancers, either as monotherapy or in combination with other agents. One such inhibitor, 6-OH-dopa, was shown to have an additive effect on BRCA-deficient cancer cells treated with PARP inhibitors, which they termed “dual” synthetic lethality [97]. Addition of RAD52 inhibitors could decrease the necessary dosage of PARP inhibitors, hopefully reducing their toxicity.

RAD52 inhibitors may also be useful outside the realm of BRCA deficiency, as depletion of RAD52 was shown to decrease tumor volume in FANCM-deficient tumors in mice. This interplay of RAD52 and FANCM was important in the repair and stability of common fragile sites, and relied on the translocase activity of FANCM rather than its association with the rest of the Fanconi anemia (FA) core complex [98]. It remains to be determined whether any of the other myriad roles of RAD52 represent a new context in which an inhibitor may be useful.

## 6. Conclusions

While RAD52 is evolutionarily conserved across species, it was long thought to be largely dispensable in humans. Recently, however, it has been implicated in a wide range of pathways throughout the cell cycle as a protector of genome stability. It is remarkable that the loss of a protein with these many functions has no observable effect on viability, but it is clearly a versatile promoter of repair of many types of damage. RAD52 has not only been found to have a role in newer pathways such as RNA-templated repair, but it may also play a larger role in HR than previously thought. Pathways such as TC-HR and TA-HRR differ as to whether or not BRCA1 and BRCA2 are required, but they agree that RAD52 is vital for the repair of DSBs at actively transcribed genes.

Unlike the classical view of synthetic lethality in DNA repair, which attributes equal consequences to the loss of each pathway, the relationship between BRCA1/2 and RAD52 appears to be quite different. BRCA deficiency on its own is a known driver of tumorigenesis, while RAD52 loss is aphenotypic. In this instance, it is possible that no single pathway of RAD52 is responsible for its synthetic lethal phenotype with BRCA1 or BRCA2, but that the cumulative effect of the disruption of so many repair pathways overwhelms the cell.

Further research is required into the newly described functions of RAD52, especially in discerning the contexts in which each pathway is relevant in the cell. A deeper understanding of the context-specific roles of RAD52 will allow for the identification of new paradigms in which RAD52 targeting may control tumor growth, even outside the realm of HR-deficient cancers.

## Figures and Tables

**Figure 1 cancers-11-01038-f001:**
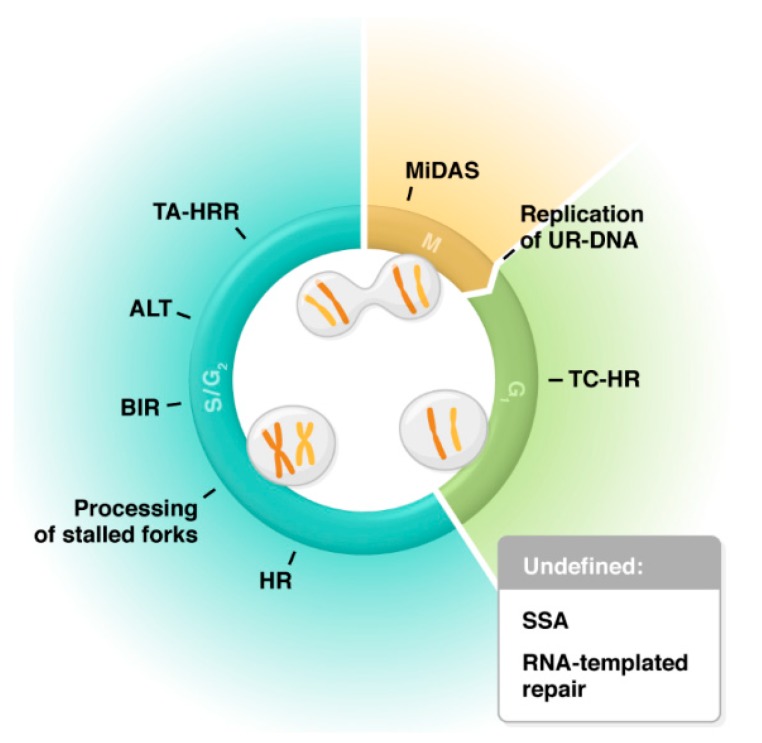
Involvement of RAD52 in various pathways of genome maintenance: TC-HR (transcription-coupled homologous recombination), HR (homologous recombination), Processing of stalled forks (MUS81-dependent fork cleavage and protection against excessive fork reversal), BIR (break-induced replication), ALT (alternative lengthening of telomeres), TA-HRR (transcription-associated homologous recombination repair), MiDAS (mitotic DNA synthesis), Replication of UR-DNA (replication of under-replicated DNA in 53BP1 nuclear bodies), SSA (single-strand annealing), and RNA-templated repair. Pathways are categorized by the cell cycle phase in which they are predominately active [24,25,26,27,28,29,30].

**Figure 2 cancers-11-01038-f002:**
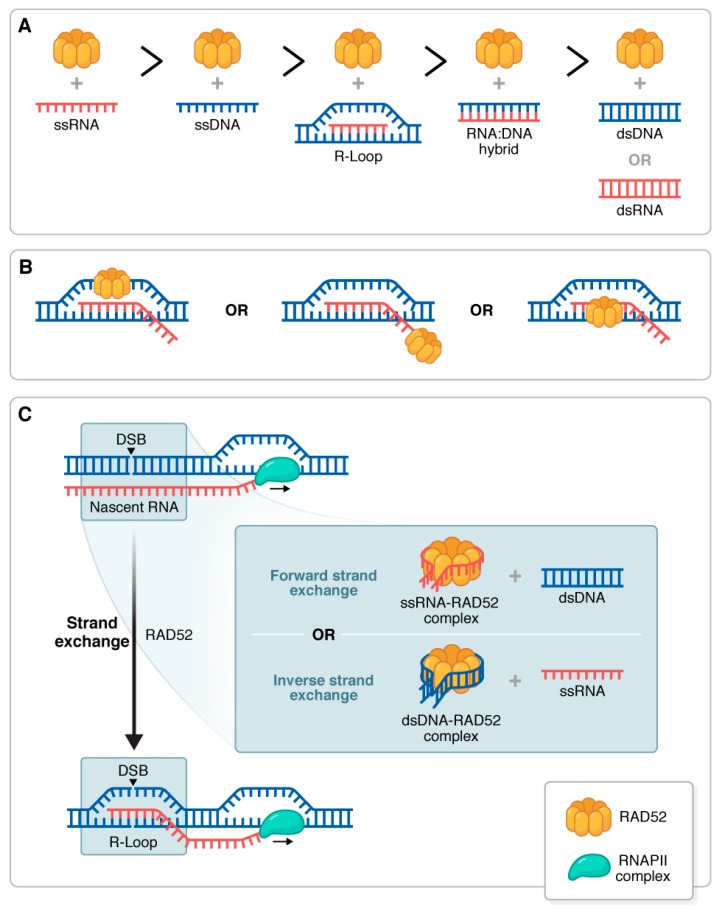
RAD52 and RNA-containing substrates. (**A**) Affinity of human RAD52 for various nucleic acid substrates adapted from mobility shift assay reported in Welty et al., 2017 [52]. (**B**) Possible modes of RAD52 binding at R-loops (to ssDNA, ssRNA, or RNA:DNA hybrid structure). (**C**) RAD52 may facilitate R-loop formation at a double-strand break by either forward or inverse strand exchange.

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
