# Peer review of "Emerging Roles of RAD52 in Genome Maintenance"

_cancers, 2019, doi:10.3390/cancers11071038_

Reviewer 1 Report

The manuscript by Jalan et al. is an interesting and comprehensive review on the current status of knowledge on RAD52 function in maintenance of genomic integrity. An especially detailed is the section focusing on R-loops, hybrid DNA:RNA structures that are currently gaining a lot of interest with regards to DNA damage response and DNA repair.

The following minor points could be considered by the authors prior publication of this work:

-       Figure 1 legend: perhaps it would be worthwhile to add a citation of the relevant work(s) pertaining to RAD52 involvement in each of the genome maintenance pathways.

-       Section 2: Visual representation of RAD52 domains as well as the known PTM sites could be provided. Also, it would be certainly of interest of the readers to elaborate a bit more on the specific PTMs of RAD52 (exact sites) and their particular roles if elucidated.

-       P. 3, line 104: It is not entirely clear, why Figure 1 is referred to at this place.

-       As already mentioned, the section on R-loops (4.3) is quite detailed and in this respect, the relevance of this section and its subsections to RAD52 (the main topic of this review) is not always given (e.g., 4.3.2. R-loops as agents of DNA damage does not refer to RAD52 at all).

- Section 5., RAD52 as a target for cancer therapy could be formally divided into 2 subsections (synthetic lethal partners and RAD52 inhibitors (strategies)). 

Author Response

Reviewer 1:

Figure 1 legend: perhaps it would be worthwhile to add a citation of the relevant work(s) pertaining to RAD52 involvement in each of the genome maintenance pathways.

We have added the relevant citations to the end of the figure legend.

Section 2: Visual representation of RAD52 domains as well as the known PTM sites could be provided. Also, it would be certainly of interest of the readers to elaborate a bit more on the specific PTMs of RAD52 (exact sites) and their particular roles if elucidated.

While the domains of RAD52 and the post-translational modifications are important, they have been discussed extensively (including visual representations) in previous reviews such as Hanamshet et al, 2016. We therefore believe that such detail is outside the scope of this review, but we have added a citation of the Hanamshet paper and have mentioned that the functions of these post-translational modifications are under active investigation.

P. 3, line 104: It is not entirely clear, why Figure 1 is referred to at this place.

We have removed the reference to Figure 1 at the above-mentioned location.

As already mentioned, the section on R-loops (4.3) is quite detailed and in this respect, the relevance of this section and its subsections to RAD52 (the main topic of this review) is not always given (e.g., 4.3.2. R-loops as agents of DNA damage does not refer to RAD52 at all).

We have added the following section to the end of Section 4.3.2:

“While these repair proteins have been shown to help prevent R-loop-induced damage, RAD52, which operates in many of the same pathways, has not appeared in any studies examining damage-causing R-loops. Nevertheless, the fact that it can bind R-loops in vitro and localize to R-loops present at break sites may indicate that a role for RAD52 at R-loop-induced damage is plausible, though this has yet to be investigated.”

We felt that it was important to represent the context-dependent damaging potential of R-loops, and as such, including this section is important even without a direct known link to RAD52.

Section 5., RAD52 as a target for cancer therapy could be formally divided into 2 subsections (synthetic lethal partners and RAD52 inhibitors (strategies)). 

Strategies to target RAD52, as well as specific inhibitors, have been well described in Hengel, Spies, and Spies, 2017. As we’re not focusing on the biochemical activity of specific inhibitors, a separate section on strategies would be quite short and out of place.

Reviewer 2 Report

In this review, the authors summarize the fundamental and recent views in the field of Rad52. Overall, the description is clear and precise. The citations are fair and appropriate. A few minor points should be addressed before publishing this excellent review.

Line 36: The BRCA1 function in HR might be dependent on the context. In the case of HR after two-ended DSBs, since the deficiency of resection in BRCA1-deficient cells is very mild, the predominant function of BRCA1 might not be in the resection step, rather in the later steps of the HR process (e.g. RPA-Rad51 exchange cooperating with BRCA2/PALB2).

Lines 72 & 121: This recent paper should be cited in terms of Rad52’s function in the ALT process.

https://www.sciencedirect.com/science/article/pii/S2211124718320771

Lines 201-202: The word “only” is better to be removed in this context because there is no way to show “direct” binding of Rad52 to R-loops other than in vitro assays.

Figure 2C: Any RNA:DNA hybrid regions associating with the ongoing transcription should sit inside the RNAPII complex.

Author Response

Reviewer 2:

Line 36: The BRCA1 function in HR might be dependent on the context. In the case of HR after two-ended DSBs, since the deficiency of resection in BRCA1-deficient cells is very mild, the predominant function of BRCA1 might not be in the resection step, rather in the later steps of the HR process (e.g. RPA-Rad51 exchange cooperating with BRCA2/PALB2).

We have added the phrase “but recent work has shown that BRCA1 may also have a function in the later steps of HR” with appropriate references.

Lines 72 & 121: This recent paper should be cited in terms of Rad52’s function in the ALT process.

https://www.sciencedirect.com/science/article/pii/S2211124718320771

We have cited this paper at the sites indicated.

Lines 201-202: The word “only” is better to be removed in this context because there is no way to show “direct” binding of Rad52 to R-loops other than in vitro assays.

We have changed “only” to “existing”

Figure 2C: Any RNA:DNA hybrid regions associating with the ongoing transcription should sit inside the RNAPII complex.

We have edited Figure 2c to include a double-strand break at the site of R-loop formation, consistent with current literature describing R-loop formation at a double-strand break. We have edited the figure legend and the reference to Fig. 2c in the paper to reflect this change.

We have also changed the structure of the nascent RNA exiting the RNAPII complex to exclude hybridization between the newly synthesized RNA and the template DNA.

Reviewer 3 Report

Despite its importance in homologous recombination in yeast, RAD52 has been recognized as a DNA repair protein whose exact roles in mammals remain unclear. However, recent studies revealing synthetic lethality of RAD52 and BRCA2 make RAD52 a therapeutic target. In the first part of the review, biochemical properties and cell cycle-associated functions of RAD52 are briefly summarized. Then, emerging roles of RAD52 in transcription-coupled homologous recombination and in transcription-associated homologous recombination repair are described in detail. Finally, synthetic lethality of RAD52 and other DNA repair proteins along with therapeutic strategies are explained. Thus, to understand the complicated roles of RAD52 in mammals, this review is very informative.

Author Response

Reviewer 3:

No changes suggested.